# Interpretable Classification via Supervised Variational Autoencoders and Differentiable Decision Trees

## Abstract

As deep learning-based classifiers are increasingly adopted in real-world applications, the importance of understanding how a particular label is chosen grows. Single decision trees are an example of a simple, interpretable classifier, but are unsuitable for use with complex, high-dimensional data. On the other hand, the variational autoencoder (VAE) is designed to learn a factored, low-dimensional representation of data, but typically encodes high-likelihood data in an intrinsically non-separable way. We introduce the *differentiable decision tree* (DDT) as a modular component of deep networks and a simple, differentiable loss function that allows for end-to-end optimization of a deep network to compress high-dimensional data for classification by a single decision tree. We also explore the power of labeled data in a supervised VAE (SVAE) with a Gaussian mixture prior, which leverages label information to produce a high-quality generative model with improved bounds on log-likelihood. We combine the SVAE with the DDT to get our *classifier+VAE* (C+VAE), which is competitive in both classification error and log-likelihood, despite optimizing both simultaneously and using a very simple encoder/decoder architecture.

## 1 Introduction

While deep learning approaches are very effective in many classification problems, interpretability of the classifier (why a particular classification was made) can be very difficult, yet critical for many applications. Decision trees are highly interpretable classifiers, so long as the data is encoded such that the classes can be easily separated. We present a *differentiable decision tree* (DDT) that we connect to a variational autoencoder (VAE) to learn an embedding of the data that the tree can classify with low expected loss. The expected loss of the DDT is differentiable, so standard gradient-based methods may be applied in training.

Since we work in a supervised learning setting, it is natural to exploit the label information when training the VAE. Thus, we employ a supervised VAE (SVAE) that uses a class-specific Gaussian mixture distribution as its prior distribution. We found that the SVAE was very effective in exploiting label information, resulting in improved log-likelihood due to separation of classes in latent space. Further, when we combined SVAE with DDT (yielding our *Classifier+VAE*, or C+VAE), we got a model that is competitive in both classification error and log-likelihood, despite optimizing both simultaneously and using a very simple encoder/decoder architecture. Further, the resultant decision tree revealed clear semantic meanings in its internal nodes.

### 1.1 Our Contributions

Our first contribution is a demonstration of the power of labeled data in autoencoding, using a simple class-based Gaussian mixture model as a VAE's prior, trained with fully labeled data. Specifically, our VAE objective function regularizes w.r.t. a class-specific Gaussian mixture model rather than to $\mathcal{N}(\mathbf{0}, I)$. Training this supervised VAE (SVAE) requires class labels, but results in better log-likelihood bounds than an unmodified VAE and excellent image generation on MNIST data, very effectively taking advantage of the class information.

Our second contribution is a *differentiable decision tree* (DDT), which allows us to differentiate the expected loss of a specific tree w.r.t. a specific distribution. This is applicable to learning embeddings, allowing us to compute the gradient of expected loss for weight updates.

Our third contribution combines SVAE with DDT to get *Classifier+VAE* (C+VAE), which learns a latent variable distribution suitable for both classification and generation simultaneously. Trained on MNIST data, C+VAE produced an encoder that is competitive in both classification error and log-likelihood, using very simple encoders and decoders.

Our final contribution is an analysis of the interpretability of a DDT trained on MNIST. Each internal node of the tree tests one of the 50 encoded dimensions when making a classification decision. We examine the values of the MNIST test data in the encoded dimensions used by the tree, analyzing the semantics of each dimension. We found that the dimensions used by the tree to discriminate correspond to meaningful macro-features learned by the encoder, and that the tree itself effectively summarizes the classification process.

The rest of this paper is organized as follows. In Section 2 we give relevant background. Then in Section 3 we describe the SVAE and present our differentiable decision tree and our combined model. Our experimental results appear in Section 4. Finally, we present related work in Section 5, and conclude in Section 6 with a discussion of future work.

## 2 BACKGROUND

We begin with a dataset of pairs, $\{(\mathbf{x}^1, y^1), \dots, (\mathbf{x}^n, y^n)\}$, where $\mathbf{x}^i \in \mathbf{X} \subset \mathbb{R}^m$ is the $i$th observation and the class label is $y^i$.

Kingma & Welling (2014) introduced the variational autoencoder (VAE) as a latent variable model for efficient maximum marginal likelihood learning. The VAE performs density estimation on $p(\mathbf{x}, \mathbf{z})$ where $\mathbf{z}$ are latent variables, to maximize the likelihood of the observed training data $\mathbf{x}$:

$$\log p(\mathbf{X}) = \sum_{i=1}^{N} \log p(\mathbf{x}^{(i)}) \ .$$

Since this marginal likelihood is difficult to work with directly for non-trivial models, instead a parametric inference model $q(\mathbf{z} \mid \mathbf{x})$ is used to optimize the variational lower bound on the marginal log-likelihood:

$$\log p(\mathbf{x}) \geq \mathbb{E}_{q(\mathbf{z}|\mathbf{x})}[\log p(\mathbf{x} \mid \mathbf{z})] - KL(q(\mathbf{z} \mid \mathbf{x})||p(\mathbf{z})) = \mathcal{L}(\mathbf{x}; \theta) \ ,$$

where $\theta$ indicates the parameters of the encoder $p$ and decoder $q$ models, and $KL(\cdot||\cdot)$ is the Kullback-Leibler divergence (Kullback & Leibler (1951)). The VAE optimizes the lower bound by reparameterizing $q(\mathbf{z} \mid \mathbf{x})$ (Kingma & Welling (2014); Rezende et al. (2014)).

The first term of $\mathcal{L}$ above corresponds to the reconstruction error of the decoder $p(\mathbf{x} \mid \mathbf{z})$, and the second term regularizes the distribution parameterized by the encoder $p(\mathbf{z} \mid \mathbf{x})$ to minimize the K-L divergence from a chosen prior distribution, usually an isotropic, centered Gaussian. The simplicity of this prior distribution has the downside of restricting the flexibility of latent variable assignment, but allows the VAE to be easily used as a decoder-based generative model as

$$p(\mathbf{x}, \mathbf{z}) = p(\mathbf{x} \mid \mathbf{z})p(\mathbf{z})$$
$$p(\mathbf{z}) = \mathcal{N}(\mathbf{z}; 0, I) \ .$$

While more sophisticated approximate posteriors have been used to improve variational inference (Salimans et al. (2015); Rezende & Mohamed (2015); Tran et al. (2015); Kingma et al. (2016); Burda et al. (2015)), for the sake of efficiency and simplicity, in our work we use only the simple approximate posterior $q(\mathbf{z} \mid \mathbf{x})$ stated above. It is straightforward to combine the more sophisticated approaches with our model.

## 3 OUR MODEL

Now we describe our model. The main contributions that this model brings in are (1) A differentiable decision tree, where we describe how to compute the expected probability distribution over predicted

labels and use this to differentiate the expected loss of the tree (which can be used to optimize learned embeddings to minimize expected loss); (2) an explicit concept of class-derived data (supervised VAE) by modifying the prior $p(\mathbf{z})$ to a mixture of Guassians; and (3) A combined VAE model using modifications (1) and (2) designed to learn a latent variable distribution suitable simultaneously for classifying and generating data (C+VAE).

## 3.1 SUPERVISED VAE

Previous work has explored the potential of the VAE's encoder to learn the data manifold for non-linear dimensionality reduction and semi-supervised classification (Kingma et al. (2014)). A difficulty arises from the VAE objective as the encoder learns to cluster the latent codes of high-probability data as close as possible to the mean of the prior distribution to minimize the K-L divergence term, $KL(q(\mathbf{z} \mid \mathbf{x}) \| p(\mathbf{z}))$, with the typical choice of unit Gaussian for the prior. This mixes data of various classes, making them difficult to separate for classification. We address this issue by changing the prior from an isotropic, unit Gaussian of the standard VAE to a mixture of unit Gaussians. Specifically, we modify the generation procedure to be class-focused, rather than assuming otherwise undifferentiated data.

To build the notion of class-distributed data into the VAE objective, we use a the prior to the following Gaussian mixture:

$$y \sim Multinomial(y; \pi)$$
$$p(\mathbf{z} \mid y) = \mathcal{N}(\mathbf{z} \mid \mu_y, \mathbf{I})$$
$$p(\mathbf{z}) = \sum_y p(\mathbf{z} \mid y)\pi(y) \tag{1}$$

where $y$ is a class label, and $\mu_y$ is the posterior mean of class $y$, and $\pi$ is a probability vector which may be pre-computed with the assumption that the labels in the training dataset are iid. As $\mu_y$ is calculated empirically from the posterior, it can be initialized to small random values for all classes and updated regularly throughout training. The objective for this VAE is

$$\mathcal{L}(\mathbf{x}; \theta) = \mathbb{E}_{q(\mathbf{z} \mid \mathbf{x})}(\log p(\mathbf{x} \mid \mathbf{z})) - KL(q(\mathbf{z} \mid \mathbf{x}) \| p(\mathbf{z})) \tag{2}$$

Since we train in a fully supervised model, each training instance is of the form $(\mathbf{x}, y)$. Thus, when $y$ is instantiated as the class variable, we get $p(\mathbf{z}) = p(\mathbf{z} \mid y)$ in Equation 1, so the KL term of Equation 2 becomes $KL(q(\mathbf{z} \mid \mathbf{x}) \| \mathcal{N}(\mathbf{z} \mid \mu_y, \mathbf{I}))$, yielding an objective of

$$\mathcal{L}'(\mathbf{x}, y; \theta) = \mathbb{E}_{q(\mathbf{z} \mid \mathbf{x})}(\log p(\mathbf{x} \mid \mathbf{z})) - KL(q(\mathbf{z} \mid \mathbf{x}) \| \mathcal{N}(\mathbf{z} \mid \mu_y, \mathbf{I})) \ . \tag{3}$$

In the remainder of this paper, we refer to a VAE trained with this objective function as the *supervised VAE (SVAE)*.

Our approach to utilizing a Gaussian mixture as a prior distribution is similar to that of Dilokthanakul et al. (2017). A key difference between their work and ours is that our use of class labels enhances training, obviating the need to marginalize over all classes to compute the K-L divergence. This helps avoid the over-regularization problem that they discuss in their paper, while achieving high sample quality in our generated images.

## 3.2 DIFFERENTIABLE DECISION TREE

The decision tree is a simple, interpretable model used for non-parametric classification. Typically, a tree is constructed by an algorithm like CART or C4.5 (Breiman et al. (1984); Quinlan (1993)) that recursively divides the dataset at each node of the tree, greedily minimizing the weighted Gini coefficient or entropy of two subsets by choosing a dividing line in one dimension. Inference is performed by walking an input from the root to a leaf according to this series of inequalities, and then assigning a class probability vector according to the leaf. Decision trees often classify well when using data with a few, richly descriptive features, and are very interpretable in their decision making processes. We are interested in learning a deep network for non-linear dimensionality reduction to allow the decision tree to classify a low-dimensional embedding of data that is normally high-dimensional and very structured. Toward this end, we utilize a probabilistic generalization of decision

trees, where each leaf returns a distribution over all classes. I.e., if instance $\mathbf{z}$ lands in leaf $\ell$ of tree $T$, then $T$ returns a distribution $P_T(y \mid \ell)$.

Fix a decision tree $T$ with leaves $L$ that takes as input instance $\mathbf{z}$ and outputs a label $y$. We observe that each leaf $\ell$ of $T$ corresponds to a region in an axis-aligned rectilinear partitioning covering the data space whose bounds are defined by the inequalities encoded in the path from the root. To compute the expected loss of $T$ on instance $\mathbf{z}$ drawn according to probability distribution $D(\mathbf{z})$, first consider one leaf $\ell$, and let $R_\ell \subset \mathbb{R}^d$ be the region of $T$'s input space that is covered by $\ell$. Then the probability that randomly drawn instance $\mathbf{z}$ falls into leaf $\ell$ is

$$P_{\mathbf{z} \sim D}(\ell \mid \mathbf{z}) = \int_{R_\ell} D(\mathbf{z}) \, d\mathbf{z} \ .$$

Then the probability that randomly drawn instance $\mathbf{z}$ is predicted to be class $y$ is

$$P_{\mathbf{z} \sim D}(y \mid \mathbf{z}) = \sum_{\ell \in L} P_T(y \mid \ell) \int_{R_\ell} D(\mathbf{z}) \, d\mathbf{z} \ .$$

This calculation is simplified if $D(\mathbf{z})$ is restricted to have diagonal covariance, i.e., $D(\mathbf{z}) = \prod_{i=1}^{d} D_i(z_i)$. Then, the integral of the PDF over $R_\ell = [r_{\ell,1}^-, r_{\ell,1}^+] \times \cdots \times [r_{\ell,d}^-, r_{\ell,d}^+]$ may be replaced with a product of the integrals in each dimension. Further, each integral may be calculated as the difference of the cumulative distribution function at the upper ($r_{\ell,i}^+$) and lower ($r_{\ell,i}^-$) bounds of the partition in each dimension $i$. The full inference may be re-written as

$$P_{\mathbf{z} \sim D}(y \mid \mathbf{z}) = \sum_{\ell \in L} \prod_{i=1}^{d} \left( \mathrm{CDF}_i(r_{\ell,i}^+) - \mathrm{CDF}_i(r_{\ell,i}^-) \right) \ ,$$

where $\mathrm{CDF}_i$ is the cumulative distribution function of $D_i(z_i)$.

Given a user-specified loss function $\mathrm{loss}_T(\mathbf{z}, y)$, our goal is to minimize the expected loss $L_T = \mathbb{E}_{\mathbf{z} \sim D}[\mathrm{loss}_T(\mathbf{z}, y)]$. In our work, we use as $D$ the distribution $q(\mathbf{z} \mid \mathbf{x}) = \mathcal{N}(\mathbf{x}; \mu_{\mathbf{x}}, \sigma_{\mathbf{x}} I)$, where $\mu_{\mathbf{x}}$ and $\sigma_{\mathbf{x}}$ are the outputs of the encoder on input $\mathbf{x}$. To perform gradient-based optimization of $L_T$, the gradient w.r.t. each parameter is calculated as

$$\frac{\partial}{\partial \mu_{\mathbf{x},i}} P_{\mathbf{z} \sim q(\mathbf{z}|\mathbf{x})}(y \mid \mathbf{z}) = \sum_{\ell \in L} \left( \mathrm{PDF}_i(r_{\ell,i}^- \mid \mu_{\mathbf{x},i}, \sigma_{\mathbf{x},i}) - \mathrm{PDF}_i(r_{\ell,i}^+ \mid \mu_{\mathbf{x},i}, \sigma_{\mathbf{x},i}) \right) \prod_{j \neq i} \left( \mathrm{CDF}_i(r_{\ell,i}^+) - \mathrm{CDF}_i(r_{\ell,i}^-) \right)$$

and

$$\frac{\partial}{\partial \sigma_{\mathbf{x},i}} P_{\mathbf{z} \sim q(\mathbf{z}|\mathbf{x})}(y \mid \mathbf{z}) = \sum_{\ell \in L} \left( (\mu_{\mathbf{x},i} - r_{\ell,i}^+) \mathrm{PDF}_i(r_{\ell,i}^+ \mid \mu_{\mathbf{x},i}, \sigma_{\mathbf{x},i}) - (\mu_{\mathbf{x},i} - r_{\ell,i}^-) \mathrm{PDF}_i(r_{\ell,i}^- \mid \mu_{\mathbf{x},i}, \sigma_{\mathbf{x},i}) \right)$$

$$\prod_{j \neq i} \left( \mathrm{CDF}_i(r_{\ell,i}^+) - \mathrm{CDF}_i(r_{\ell,i}^-) \right) \ ,$$

where $\mathrm{PDF}_i(r \mid \mu_{\mathbf{x},i}, \sigma_{\mathbf{x},i})$ is the value of the Gaussian PDF of dimension $i$ evaluated at $r$.

This allows optimization of the distribution parameters for maximum likelihood w.r.t. an existing decision tree $T$. Thus, an embedding of the data may be learned in an EM-style manner, alternately learning a tree on the embedding produced by the parameters of a deep encoder and optimizing the embedding parameters to better fit the class-based partitioning induced by the learned decision tree.

### 3.3 OUR COMBINED MODEL

The supervised, Gaussian-mixture-based VAE and decision tree inference can be used with a VAE model to both classify **and** reconstruct data from the encoded parameters of its latent variable distribution. Although an embedding could be learned by only optimizing the classification accuracy of the decision tree, the additional reconstruction objective ensures that the learned representation is non-arbitrary and contains more than just class information for downstream use.

Our new architecture C+VAE (Classifier+VAE) uses a deep encoder network to parameterize a Gaussian distribution, which is then used as the input for classifying with the DDT and to reconstruct

the encoded data with a deep decoder network. Generally, the combined modifications can also be applied to existing VAE architectures when label information is available.

The C+VAE training procedure begins by randomly initializing the encoder/decoder parameters and encoding the training data to initialize the decision tree and aggregate posterior class means. Training then proceeds by running several epochs of gradient updates before re-training the decision tree and updating the aggregate posterior class means until the model converges.

The optimization function of our combined model consists of a linear combination of the objective of the supervised VAE and the expected error of the current decision tree $T$. However, since the effect of the DDT gradient is to separate the class means and the supervised VAE K-L divergence term measures w.r.t. these movable class-based means (rather than the distance from the mean of $p(\mathbf{z})$), the parameters learned by the encoder could diverge, driving the class means arbitrarily far from the origin. Thus, an additional regularizing $L_2^2$-loss is imposed on the encoded $\mu$ posterior value to keep the class means from "drifting" from the origin and encourage the model to learn common factors of variation between classes. We observed experimentally that this additional drift loss term increases the training stability and classification performance of the model. The modified VAE objective of the C+VAE to be minimized is

$$f(\mathbf{x}, y; \theta) = -\mathcal{L}'(\mathbf{x}, y; \theta) + \gamma L_T + \lambda \|\mu_x\|_2^2 \ . \tag{4}$$

## 4 EXPERIMENTS

Our experiments are designed to empirically study the following claims:

1. The supervised VAE very effectively takes advantage of class labels to improve generative performance.

2. The C+VAE classifies competitively with other tree-based embedding methods while simultaneously maintaining a generative model competitive with the literature.

3. The differentiable decision tree is an interpretable classifier that, when used in C+VAE, can learn the semantics of the macro-features learned by the underlying VAE.

The MNIST dataset was used for all experiments, as it is widely understood and commonly used for both classification and generation tasks. We applied the C+VAE modifications to a standard VAE (Kingma & Welling (2014)) with two-layer MLPs of 500 hidden units as encoder and decoder models and a 50-dimensional latent variable $\mathbf{z}$ without importance sampling or an autoregressive prior. The CART algorithm as made available in scikit-learn (Pedregosa et al. (2011)) was used to train the decision tree. This was regularized by annealing the maximum depth of the decision tree from 1 to 8 as training proceeded, incrementing every 15 epochs, and by setting the minimum proportion of samples in a leaf to be 2% of the training set. Unless otherwise noted, we used $\gamma = 1000$ and $\lambda = 0.1$ in the objective function of the C+VAE (Equation (4)), and $n = 5$ epochs of gradient steps between each update of both the decision tree and the aggregate posterior class means. Adam (Kingma & Ba (2014)) was used for optimization and the data was not pre-processed or augmented.

Table 1 lists the classification performance of a number of tree-based and VAE-based models. The M1:SVAE+CART model trains the supervised VAE to convergence, and then trains a standard decision tree with CART to classify its latent code in the style of M1 (Kingma et al. (2014)). The intent is to highlight the effect of training without the backpropagated classification loss from the DDT. C+VAE sans reconstruction zeros the reconstruction loss term of the objective function to highlight the effect of training a model that only learns an embedding suitable for classification with the DDT. The boundary tree (BT) with embedding is from Zoran et al. (2017), M1+M2 is from Kingma et al. (2014), and the Ladder Network is from Rasmus et al. (2015).

### 4.1 EVALUATING THE SUPERVISED VAE

We first evaluate the efficacy of leveraging label data in a supervised VAE in generation. I.e., the effect on generation of making the prior distribution a Gaussian mixture and taking advantage of class label information. This is equivalent to using C+VAE with $\gamma = \lambda = 0$ in Equation (4). The flexibility of a Gaussian mixture and the fact that the data is clearly multi-modal both contribute to

Table 1: Classification error for various fully supervised tree- and VAE-based models.

| Model | Error |
|---|---|
| M1:SVAE+CART | 37.09% |
| C+VAE sans reconstruction | 7.30% |
| C+VAE | 1.98% |
| BT w/embedding | 1.85% |
| M1+M2 | 0.96% |
| Ladder Network | 0.57% |

the SVAE log-likelihood of $-102.77$, which is better than the log-likelihood of $-109.56$ using our implementation of the VAE of Kingma & Welling (2014), which uses an unmodified Gaussian prior. We expect this difference to be the result of using a flexible prior that is more faithful to the true prior. This flexibility is similar to that seen in techniques like normalizing flows (Rezende & Mohamed (2015)), but modifies the prior rather than the posterior and uses the additional information provided by label information, rather than adding additional computation.

## 4.2 EVALUATING THE C+VAE

We next empirically evaluate C+VAE for both classification performance and generative ability. As a baseline, we first examine how well a standard (non-differentiable) decision tree from CART can classify when the data is encoded by a supervised VAE (but without any error feedback: $\gamma = \lambda = 0$). This is similar to M1 from Kingma et al. (2014) with a different VAE. In Table 1, row M1:SVAE+CART shows that without the error feedback from the tree, it is unlikely that the embedding will be useful in classification by a decision tree. This motivates our use of the DDT.

To test the benefit of reconstruction in learning an embedding that can be classified well, we ran a test in which we switched off the reconstruction error feedback in learning. I.e., we removed the first term of Equation (3). In Table 1, row C+VAE sans reconstruction shows a significant improvement in classification error over M1:SVAE+CART, but still quite high.

Row C+VAE in Table 1 shows our combined method's performance with $\gamma = 1000$ and $\lambda = 0.1$. We see a large improvement in classification error over C+VAE sans reconstruction, demonstrating the importance of both types of feedback in training. While C+VAE's classification performance is worse than results from the literature, it's still competitive, despite simultaneously optimizing both classification and log-likelihood. Also, we note that C+VAE's log-likelihood of $-110.12$ is comparable to the $-109.56$ from our implementation of the VAE of Kingma & Welling (2014), which uses the same encoder-decoder pair as C+VAE. A more powerful encoder or the use of more recent techniques (e.g., normalizing flows, importance weighting, etc.) could conceivably improve both error and log-likelihood even further.

Figure 7 in the appendix presents sample MNIST digits generated by C+VAE. Each set of digits is generated from one of the empirical aggregated class means.

## 4.3 EVALUATING THE INTERPRETABILITY OF THE DDT

The final decision tree learned by the C+VAE is shown in Figure 1 (a landscape version of the same tree is in Figure 8 in the appendix). This tree performs feature selection over the 50 available latent dimensions, using only 8 to classify with 98.02% accuracy with one dimension (21) used to split twice. We were able to leverage the simplicity of the decision tree to assign meaning to the latent dimensions used by the tree to classify inputs. Each node divides inputs according to a threshold value in a single dimension, which corresponds to detecting the most salient macro-feature that distinguishes the divided subsets.

Dimension 21 of the latent code is the macro-feature used by the decision tree to discriminate between digits '6' and '0', as well as '4' and '7'. Figure 2 visualizes the macro-feature corresponding to dimension 21. Specifically, the top image (starting with '6') was generated by fixing the other 49 dimensions to be the values of $\mu_6$ and varying the value of dimension 21 in even steps from $-1$ to 2. The bottom image was generated the same way, with the other 49 dimensions initialized to

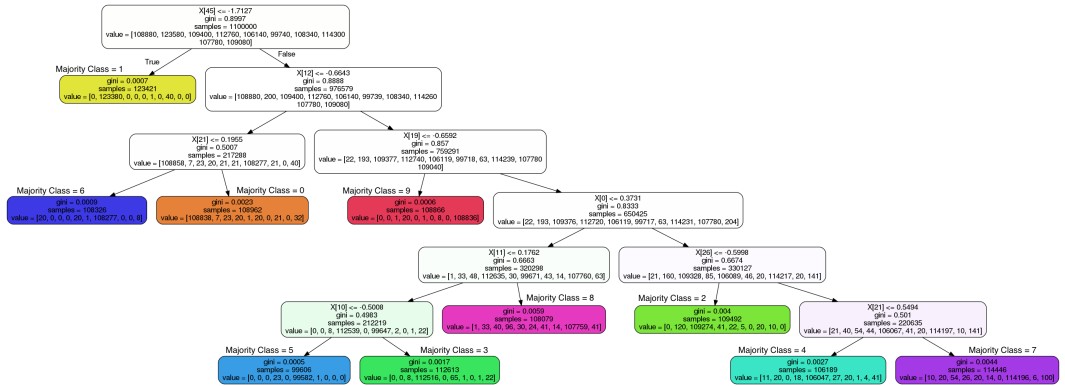

Figure 1: Decision tree learned by C+VAE on the MNIST dataset. Each internal node indicates which of the 50 latent dimensions is tested. The bottom line of each node is the number of generated instances of each class that fell to that node. The majority class of each leaf is printed next to that leaf.

values from $\mu_4$. In both image sequences, we see that a high value of dimension 21 emphasizes the macro-feature of a flat top bar of a digit whereas a low value removes it.

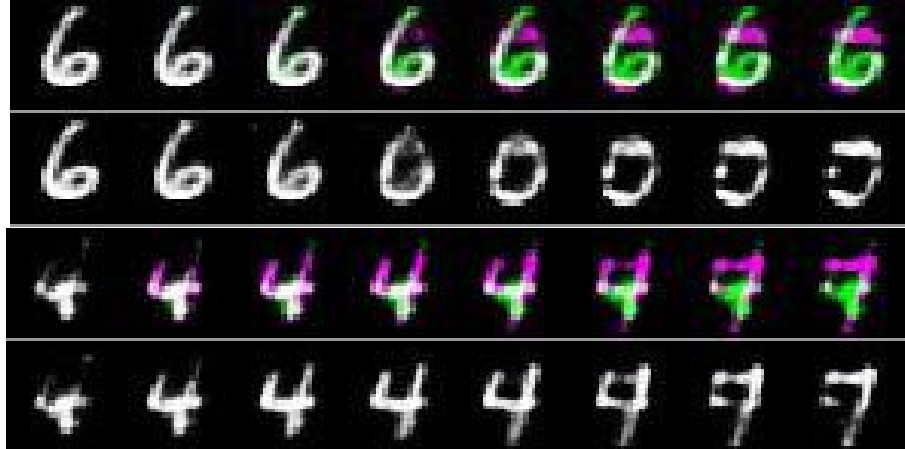

Figure 2: Varying dimension 21, other dimensions set to $\mu_6$ (top) and $\mu_4$ (bottom). From left to right, green indicates pixels which are fading, while magenta indicates pixels that are intensifying.

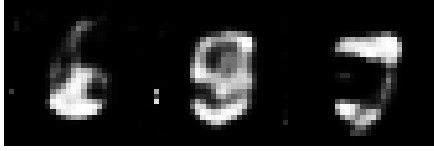

Figure 3: Varying dimension 21, other dimensions set to mean of class means.

To illustrate the effect of varying dimension 21 independently of a specific class mean, we generate Figure 3 by an identical process, but fix the other 49 dimensions to be the mean of all 10 class means. The center image is the mean of all 50 dimensions, and is provided for contrast. The left and right images show the effect of a low or high value of dimension 21 on this 'average digit'. The clearest effect of this variation is that the flat top macro-feature is present when this value is high, and absent when it is low, just as in Figure 2.

We generate Figures 4, 5, and 6 by the same process, varying dimensions 10, 26, and 45, respectively. The decision tree uses dimension 0 to separate digits '3', '5', and '8' from digits '2', '4', and '7'. Figure 4 shows that a low value of dimension 0 correlates strongly with rounded digits, while a high value creates an emphasized right side and a more angular appearance. Figure 5 shows the effect of varying dimension 26, used by the tree to separate '2' from '4' and '7'. The most notable impact of a low dimension 26 is the exaggerated lower-left corner, which is absent when that dimension is high. Figure 6 shows the effect of varying dimension 45. This is the first dimension used by the decision tree, to split '1' from the other nine classes. A clear vertical line near the center of the digit is emphasized by low values of dimension 1. Latent codes with high values of dimension 1, a feature common to the other nine classes, lack that central vertical line.

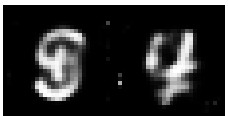

Figure 4: Varying dimension 0, other dimensions set to mean of class means.

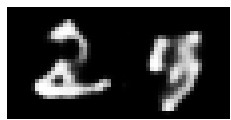

Figure 5: Varying dimension 26, other dimensions set to mean of class means.

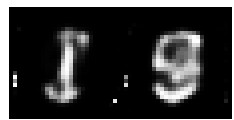

Figure 6: Varying dimension 45, other dimensions set to mean of class means.

## 5 RELATED WORK

**Differentiable Trees** Previous work that uses deep networks for representation learning with decision trees is the Deep Neural Decision Forests of Kontschieder et al. (2015), which stochastically make routing decisions through a decision tree according to the outputs of a deep convolutional network. The setup achieved good performance at its classification task, but it is not clear how to interpret the proposed classification process, especially when more than one tree is combined into a forest classifier. As a method of making the decision tree differentiable, our proposed inference method of integrating a probability distribution over the decision regions of the tree is a novel approach.

Another tree-based method uses differentiable boundary trees to learn an embedding suitable for k-nearest neighbor classification (Zoran et al. (2017)). The learned representation allows a small, interpretable boundary tree to classify effectively, similar to our technique. The classification accuracy of the technique marginally outperforms our combined model, but the C+VAE also acts as a generative model and does not suffer from the significant complexity of having to use dynamically constructed computation graphs.

**VAE learning** Other work in classifying the latent codes produced by a VAE includes Kingma et al. (2014), whose M1 semi-supervised model learns to classify from the latent embedding similarly to our combined classifier. However, M1 trains the discriminator separately from the VAE and lacks interpretability as the class separation is performed solely by a black-box discriminator. The M2 model is similar to the supervised VAE, but doesn't change the VAE prior.

Dilokthanakul et al. (2017) present a Gaussian Mixture Variational Autoencoder to learn a class-focused latent representation. Our work assumes a supervised, rather than the GMVAE's unsupervised environment. This allows the classifying modification to the VAE framework to remain simpler and more interpretable, as well as more tractable optimization.

## 6 FUTURE WORK

Future work includes applying our approach to other data sets such as CIFAR-10, and using more powerful encoders and decoders to see how performance is affected. We will also look into extending our approach to handle unlabeled data in applications such as semi-supervised learning and clustering.

ACKNOWLEDGEMENTS

We gratefully acknowledge the support of NVIDIA Corporation with the donation of the Titan X Pascal GPU used for this research.

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

## A    GENERATED MNIST IMAGES

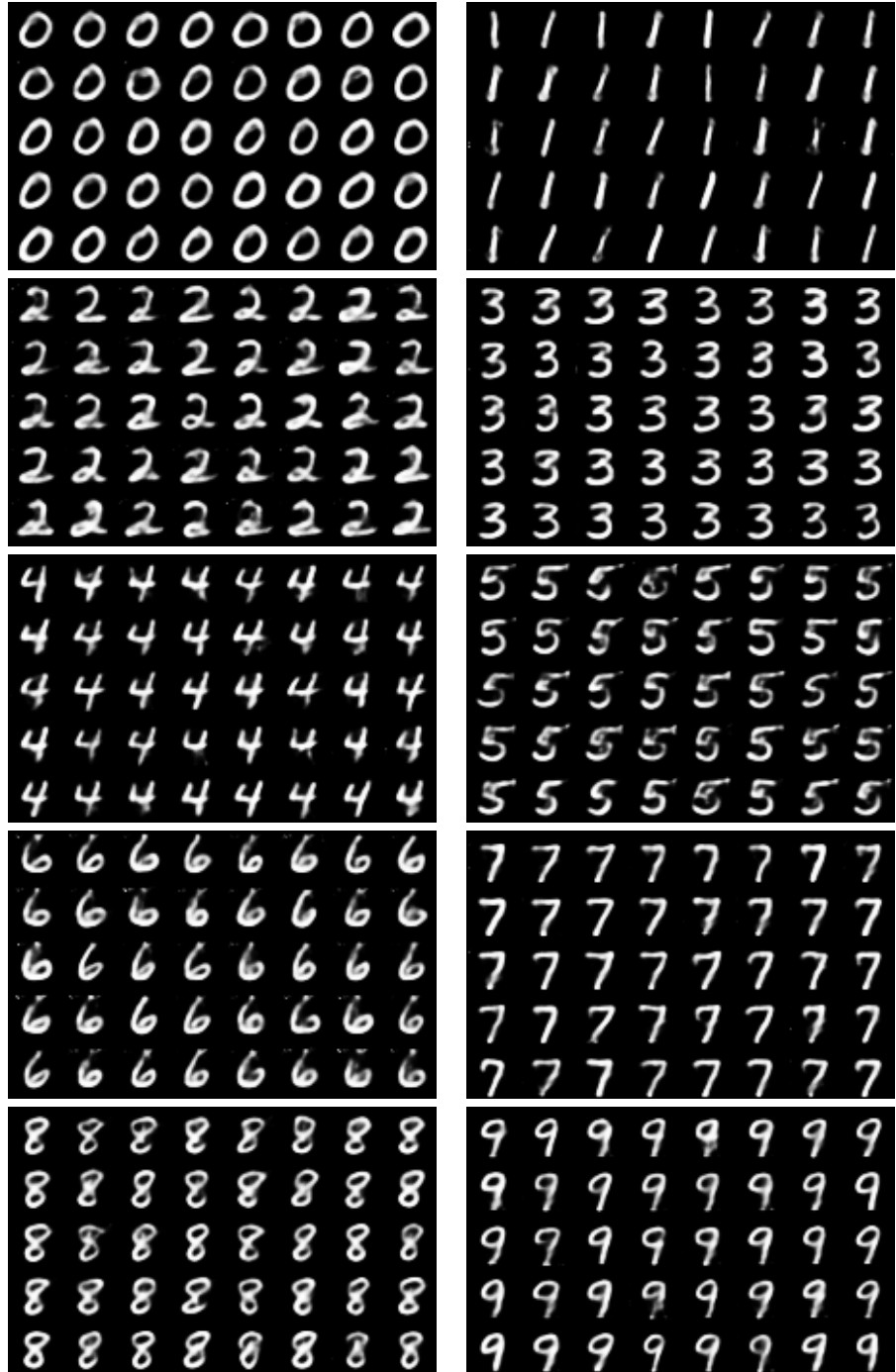

Figure 7: Sample MNIST digits generated by C+VAE. Each set of digits is generated from one class prior, but the mixture may be sampled from with one extra step.

# B  LANDSCAPE VERSION OF DECISION TREE

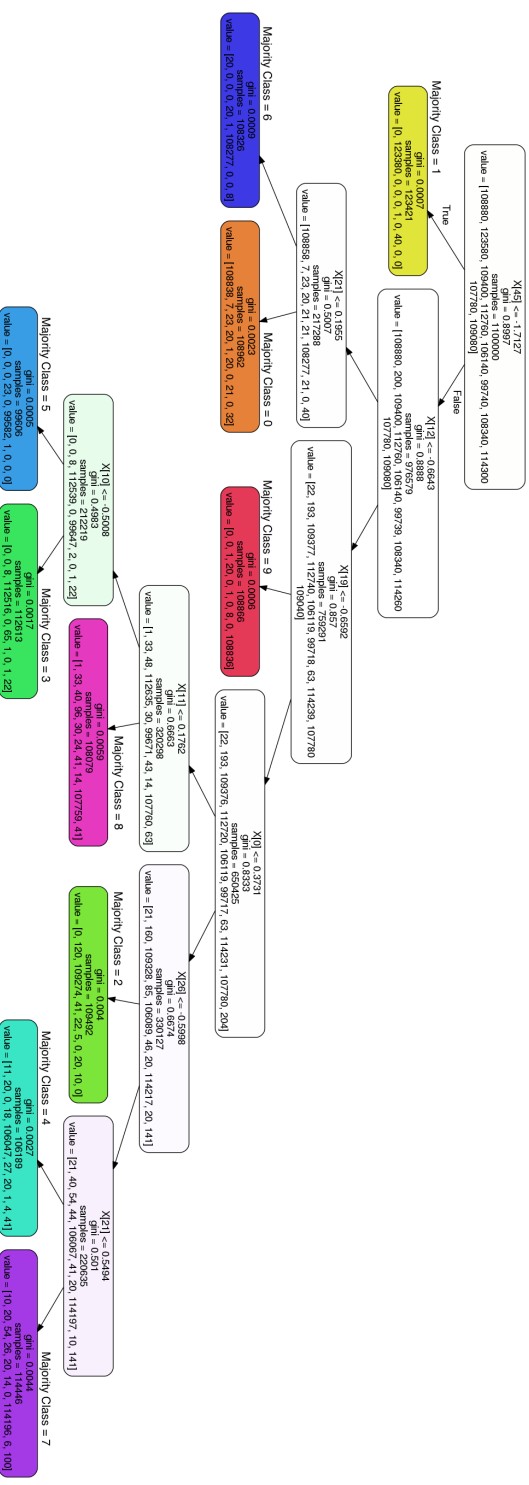

Figure 8: Landscape version of decision tree learned by C+VAE.

