# OpenReview forum: "Interpretable Classification via Supervised Variational Autoencoders and Differentiable Decision Trees"
_ICLR.cc/2018/Conference — Reject_

### Official Review · AnonReviewer2 · 2017-11-16
**Review: Interesting hybrid model, but weak experiments (MNIST only)**

**Rating:** 3
**Confidence:** 5

**Review:**


Summary

This paper proposes a hybrid model (C+VAE)---a variational autoencoder (VAE) composed with a differentiable decision tree (DDT)---and an accompanying training scheme.  Firstly, the prior is specified as a mixture distribution with one component per class (SVAE).  During training, the ELBO’s KL term uses the component that corresponds to the known label.  Secondly, the DDT’s leaves are parametrized with the encoder distribution q(z|x), and thus gradient information flows back through the DDT into the posterior approximations in order to make them more discriminative.  Lastly, the VAE and DDT are trained together by alternating optimization of each component (plus a ridge penalty on the decoder means).  Experiments are performed on MNIST, demonstrating tree classification performance, (supervised) neg. log likelihood performance, and latent space interpretability via the DDT.


Evaluation

Pros:  Giving the VAE discriminative capabilities is an interesting line of research, and this paper provides another take on tree-based VAEs, which are challenging to define given the discrete nature of the former and continuous nature of the latter.  Thus, I applaud the authors for combining the two in a way that admits efficient training.  Moreover, I like the qualitative experiment (Figure 2) in which the tree is used to vary a latent dimension to change the digit’s class.  I can see this being used for dataset augmentation or adversarial example generation, for instance.

Cons:  An indefensible flaw in the work is that the model is evaluated on only MNIST.  As there is no strong theory in the paper, this limited experimental evaluation is reason enough for rejection.  Yet, moreover, the negative log likelihood comparison (Table 2) is not an informative comparison, as it speaks only to the power of adding supervision.  Lastly, I do not think the interpretability provided by the decision tree is as great as the authors seem to claim.  Decision trees provide rich and interpretable structure only when each input feature has clear semantics.  However, in this case, the latent space is being used as input to the tree.  As the decision tree, then, is merely learning hard, class-based partitioning rules for the latent space, I do not see how the tree is representing anything especially revealing.  Taking Figure 2 as an example (which I do like the end result of), I could generate similar results with a black-box classifier by using gradients to perturb the latent ‘4’ mean into a latent ‘7’ mean (a la DeepDream).  I could then identify the influential dimension(s) by taking the largest absolute values in the gradient vector.  Maybe there is another use case in which a decision tree is superior; I’m just saying Section 4.3 doesn’t convince me to the extent that was promised earlier in the paper (and by the title).

Comment:  It's easier to make a latent variable model interpretable when the latent variables are given clear semantics in the model definition, in my opinion.  Otherwise, the semantics of the latent space become too entangled.  Could you, somehow, force the tree to encode an identifiable attribute at each node, which would then force that attribute to be encoded in a certain dimension of latent space?

---

### Official Review · AnonReviewer1 · 2017-11-24
**Some comments on the evaluation and results**

**Rating:** 4
**Confidence:** 4

**Review:**

The paper tries to build an interpretable and accurate classifier via stacking a supervised VAE (SVAE) and a differentiable decision tree (DTT). The problem is important and interesting. The authors list the contributions of each part but it seems that only the final contribution, i.e. analysis of the interpretability, is interesting and should be further extended and emphasized. Here with the detailed comments.

1. I think Table 2 does not make sense at all. This is not only because the authors use the label information but also because the authors compare different quantities. The the previous methods evaluate log p(x) while the proposed method evaluates log p(x, y) which should be much lower as the proposed method potentially trains a separated model for each class of the x for evaluation.

2. The generation results of the SVAE shown in Figure 7 in Appendix A seem strange as the diversity of the samples is much less than those from the vanilla VAEs. Could the authors explain this mode collapse phenomenon?

3. The results in Table 1 are not interesting.  It is most useful to interpret the state-of-the-art classifier while the results of the proposed methods are far from the state-of-the-art even on such simple MNIST dataset.

4. The most interesting results of this paper are shown in Figure 1. However, I think the results on the interpretability should be further extended. Several questions are as follows:

Why other dimensions are not so interpretable, compared with 21?

Can we also interpret a VAE given labels by varying each dimension of the latent variables without jointly training a DTT? I personally think some of the dimensions of the latent variables of the vanilla VAEs can also be interpreted via interpolation in each dimension.

Can these results be generalized to other datasets, consisting of natural images?

Overall, this paper is below the acceptance threshold.

---

### Official Review · AnonReviewer3 · 2017-11-26
**Interpretable classifier via deep learning is an important topic, but the work in this paper is not a substantial contribution.**

**Rating:** 5
**Confidence:** 4

**Review:**

This paper addresses a method of building an interpretable model for classification, where two key ingredients are (1) supervised variational autoencoder and (2) differentiable decision tree. Recently one important line of research is to build interpretable models which have more modeling capacity while maintaining interpretability, over existing models such as linear models or decision trees. In this sense, the current work is timely research. A few contributions are claimed in this paper: (1) differentiable decision tree which allows for gradient-based optimization; (2) supervised VAE where class-specific Gaussian prior is used for the probabilistic decoder in the VAE; (3) combination of these two models. Regarding the differentiable decision tree, I am not an expert in decision tree. However, I understand that there have been various work on probabilistic decision tree, Bayesian decision tree, and Mondrian tree. More literature survey might be needed to pin-point what's new and what's common with previous work. Regarding the supervised VAE, the term "supervised VAE" is misleading. To me, the current model is nothing but VAE with class-specific Gaussian prior. (3)  Regarding the combination of supervised VAE and DDT, it would be much better to show us a graphical illustration of the model to improve the readability. I see the encoder is common for both the decoder and DDT. However, it is not clear how DDT is coupled with the encoder. It seems that DDT takes the output of the encoder as input but the output of DDT is not coupled with VAE.

---

### Author Response · Authors · 2017-11-30
**Correction of calculations of log-likelihood results**

We recently discovered a numerical error of calculation of KL-divergence, which impacted final calculation of log-likelihood of our models SVAE and C+VAE.  Our updated bounds for log-likelihood are -102.77 for SVAE and -110.12 for C+VAE.  (Classification results were unchanged.)

In the new version we plan to upload soon, we also updated the discussion to reflect that, while our models no longer greatly improve over more complex, state-of-the-art models in terms of log-likelihood, SVAE still improves over an unmodified VAE (which uses the same encoder-decoder pair that we use), and C+VAE is comparable to an unmodified VAE when simultaneously optimizing for both classification and generative performance.

---

### Author Response · Authors · 2017-12-07
**Thanks to the reviewers for the helpful comments**

We greatly appreciate the detailed feedback from the reviewers, and will look into refocusing our paper on the interpretability aspects.

We updated the pdf to fix the bug mentioned in our earlier comment, but made no other changes at this time, pending the refocusing described above.

---

### Decision · Program_Chairs · 2018-01-29
**ICLR 2018 Conference Acceptance Decision**

**Decision:**

Reject

**Comment:**

The paper proposes a new model called differential decision tree which captures the benefits of decision trees and VAEs. They evaluate the method only on the MNIST dataset. The reviewers thus rightly complain that the evaluation is thus insufficient and one also questions its technical novelty.